# Beyond Monoliths: Expert Orchestration for More Capable, Democratic, and Safe Large Language Models

## Abstract

This position paper argues that the prevailing trajectory toward ever larger, more expensive generalist foundation models controlled by a handful of big companies limits innovation and constrains progress. We challenge this approach by advocating for an "Expert Orchestration" framework as a superior alternative that democratizes LLM advancement. Our proposed framework intelligently selects from thousands of existing models based on query requirements and decomposition, focusing on identifying what models do well rather than how they work internally. Independent "judge" models assess various models' capabilities across dimensions that matter to users, while "router" systems direct queries to the most appropriate specialists within an approved set. This approach delivers superior performance by leveraging targeted expertise rather than forcing costly generalist models to address all user requirements. The expert orchestration paradigm represents a significant advancement in LLM capability by enhancing transparency, control, alignment, and safety through model selection while fostering a more democratic ecosystem.

## 1 Introduction

The field of artificial intelligence has witnessed remarkable progress, largely driven by advancements in large language models (LLMs). Currently, users predominantly rely on monolithic frontier LLMs for queries and tasks. When these models fall short by producing hallucinations [Simhi et al., 2025, Zhang et al., 2023], showing bias [Gallegos et al., 2024], or lacking specialized knowledge [Kandpal et al., 2022] the typical response from both developers and users has been to attempt to "fix" these shortcomings through techniques like prompt engineering, RLHF, vector steering, or parameter-efficient fine-tuning. These interventions are not only time and compute intensive to implement, but they also cause "whack-a-mole" side effects

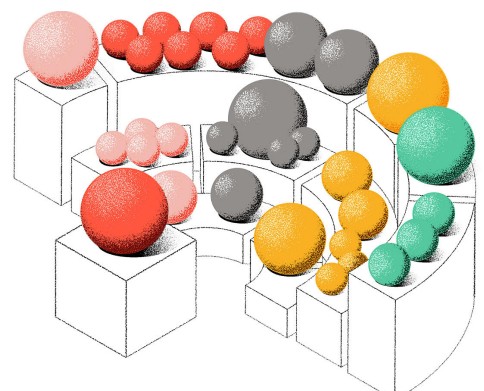

Figure 1: Conceptual view of a router that decomposes a user query then utilizes the most appropriate specialist and generalist models to process it.

where they may compromise the performance of the model under consideration, as shown by researchers in studies for fine-tuning [Shumailov et al., 2024], steering [Stickland et al., 2024], RLHF [Kirk et al., 2023], and poor prompt selection [Cao et al., 2024].

We believe that it is fundamentally intractable [Varoquaux et al., 2025] to develop a single model capable of optimal performance across all possible tasks. Hence, this paper argues against the prevailing approach of building ever-larger generalist models. Patching individual models to perform well across all domains is akin to forcing a general practitioner to perform brain surgery rather than deferring the task to a neurosurgeon. Just as humans instinctively consult different experts based on their specific competencies, we should focus on identifying and leveraging the strengths of specialized models. Such specialization not only mirrors natural expertise distribution in human communities, but also offers a more effective path to addressing the limitations of current AI systems.

There are thousands of specialized models currently available on platforms like HuggingFace [Horwitz et al., 2025]. By evaluating their strengths, we can leverage the diversity and specialization of these models. While both generalist and specialist models may require improvements, fixing specialist models is fundamentally more tractable for several reasons. First, specialist models operate in constrained domains with clearer evaluation metrics and more easily established ground truth. Second, the narrower input space dramatically reduces the testing matrix needed to ensure quality. Third, specialized human expertise can be more effectively applied to the limited domain. Finally, when specialists are improved, the benefits immediately propagate through the orchestration system without disrupting other domains unlike monolithic models where fixes for one domain often cause regressions in others due to parameter interference [Shao and Feng, 2022, Saunders and DeNeefe, 2024, Xu et al., 2020]. At the same time, [Zaharia et al., 2024] argues that state-of-the-art AI performance is increasingly driven not by scaling individual models, but by assembling compound systems composed of multiple coordinated components.

**We posit that improvements in monolithic models aiming to handle all tasks is unsustainable. Rather, we propose a paradigm shift towards a framework we term *expert orchestration*, which is comprised of specialized components: *Judges* that evaluate model capabilities across dimensions that matter to users (factuality, domain expertise, ethics, creativity, etc.), and *Routers* that direct user queries to the most appropriate model(s) in a set of specialist & generalist models, based on user preferences. This approach improves control and monitoring, delivering superior answers at lower average cost creating a capable, democratic, and safe ecosystem.**

Below, we outline limitations of the current landscape (Section 2), why interdisciplinary frameworks argue for change (Section 3), describe the expert orchestration framework (Section 4), argue it enhances LLM utility (Section 5), while acknowledging open research questions (Section 6) and alternative viewpoints exist (Section 7). Finally we urge adoption of this promising approach to secure a safer AI future (Section 8).

## 2 The Problem with Concentrating AI

Wu [2011] describes the recurring cycle, where information industries— such as radio and the internet— begin in a period of innovation but become consolidated by monopolies, which may suppress competition and innovation [Aghion et al., 2023]. We highlight the growing risks of similar AI concentration, and call for scrutiny of democratic alternatives by the research community.

### 2.1 Market Dynamics

The economics of frontier AI development are increasingly shaped by powerful market incentives that favor "winner-take-all" outcomes, where a small number of dominant players capture the lion's share of profits and influence. These companies operate with the expectation that the developers of the most capable generalist LLMs will capture the vast majority of the market, pursuing this consolidation with the potential to concentrate trillions of dollars and substantial deal-making power within a limited number of corporations.

This concentration emerges from multiple reinforcing factors. The enormous compute requirements for training frontier-scale generalist models effectively exclude most organizations from creating competitive alternatives, leading to concentration among a few well-resourced companies. The Atlantic Council highlights these economic tendencies toward winner-take-all dynamics due to significant training costs, while INET Economics points to growing fixed costs of pre-training, increasing scarcity of high-quality training data, and intense competition for talent as key drivers of market concentration [INET Economics, 2025].

Even when smaller organizations successfully develop highly specialized models that excel in specific domains, the dominant user interaction paradigm—which drives users toward single 'do-everything' interfaces—prevents these specialized models from gaining market traction despite their superior performance in their niche. This creates a two-pronged barrier: high resource requirements for generalist competition and limited market access for specialized excellence.

These market dynamics create concerning misalignments regarding safety. Frontier companies, while often expressing commitment to safety, face real incentives to under-evaluate and under-report potential risks to expedite model releases. This rush to market is driven by intense competition and the desire to capture market share in the perceived "winner-take-all" dynamic. The DarkBench paper [Kran et al., 2025] demonstrates that models misrepresent their own capabilities and advantages over competitors, further complicating accurate risk assessment.

The underlying competitive dynamics often favor rapid capability advancement and market share capture over meticulous safety assurance [Martian, 2025]. For companies selling access to increasingly powerful models, the motivation may be weak to dedicate significant resources to in-depth safety research that could slow capability advancements.

This concentration raises broader concerns about societal equity and benefit distribution [Americans for Responsible Innovation, 2025]. The Economic Policy Panel warns of systemic risks and potential inequality from such market dominance [Economic Policy, 2025], while New America argues that select technology giants are already leveraging their resources to monopolize the industry, effectively excluding competitors [Inequality.org, 2025]. Beyond economics, the concentration of such immense power in limited entities contradicts fundamental democratic principles [New America, 2025] and creates what Plus Info describes as broader dangers including potential economic and social disruptions and existential risks [AI Plus Info, 2025].

Expert orchestration addresses aspects of these market failures by lowering the resource threshold for meaningful contribution to the AI ecosystem and by creating a framework that naturally incorporates and highlights specialized excellence, regardless of the model creator's scale or resources.

## 2.2 Technical Challenges of monoliths

**Limited User Insight into LLM "Thinking" Characteristics.** Beyond the technical correctness, users are increasingly concerned with a range of underlying "thinking" characteristics. These include legality, morality, the absence of hallucinations, and the lack of gender or other biases. Currently, users have limited means to effectively communicate these criteria to LLMs and possess very limited ability to evaluate how well these models align with their desired thinking characteristics.

Frontier LLMs, however, often present themselves as being universally capable, without any clear differentiation regarding underlying thinking characteristics. While users can gain some limited control over these characteristics by their choice of LLM, and by employing specialized prompts, the actual impact and reliability of these methods remain uncertain.

This limitation is widely recognized in the alignment literature, where recent work emphasizes the importance of user-steerable LLMs and controllable generation. For instance, Bai et al. [2022] introduce Helpful and Harmless Assistant (HH-RLHF), where preferences are directly integrated into model behavior via human feedback loops and fine-tuning procedures.

Similarly, OpenAI's InstructGPT paper [Ouyang et al., 2022] shows that aligning LLMs with user intent through instruction-following dramatically improves user satisfaction and safety. However, these efforts are largely global alignment efforts so users do not have fine-grained, per-query control.

Users deserve more direct insight and specific control over the "thinking" characteristics of LLM behavior. None of the above methods match the explicit and modular control enabled by expert orchestration where each thinking characteristic (e.g., legality, bias, hallucinations) is explicitly evaluated and can be chosen by the user per query.

**Monolithic Systems Are Less Controllable.** While specialized models and frameworks that enable calling multiple models as tools do exist, ease of use considerations often lead most users to opt for a single LLM, with its inherent strengths and weaknesses, for all their queries.

This single LLM presents as a "monolith" that is sufficiently proficient across all query types. While this might hold true on average, it is demonstrably false at the individual query level. For many

queries, other LLMs, potentially with specialist abilities directly relevant to the query, would be more suitable. Alternatively, a query might be simple enough (e.g., a basic arithmetic problem) that invoking a frontier model represents a wasteful expenditure of resources: money, time, and electricity.

The shift from monoliths to components also mirrors the move in NLP and CV towards modular sparse systems [Riquelme et al., 2021] and BASE Layers [Lewis et al., 2021], which show that task-specific experts outperform generalist models at lower cost and complexity.

The Modular Deep Learning paper [Pfeiffer et al., 2023] says "It remains unclear how to develop models that specialize towards multiple tasks without incurring negative interference and that generalize systematically to non-identically distributed tasks". The paper promotes modular deep learning as a potential partial solution to these challenges.

# 3    Why Specialization Works: Lessons from Other Fields

Our position on expert orchestration is grounded in theoretical frameworks including economics, cognitive science, democratic theory and biology. These perspectives collectively demonstrate that monolithic LLMs face fundamental — not temporary — limitations, and confirm distributed expertise as a more principled architectural approach.

**Economic Theories of Distributed Knowledge and Market Structure.**    Friedrich Hayek's seminal work on distributed knowledge [Hayek, 1945] provides a powerful economic framework supporting expert orchestration. Hayek argued that knowledge in society exists as "*dispersed bits of incomplete and frequently contradictory knowledge which all the separate individuals possess,*" never in "concentrated or integrated form" in any single mind. This impossibility of centralizing all knowledge leads to the superiority of market mechanisms over central planning: markets function as information processors that coordinate distributed expertise through price signals.

Adam Smith's theory of the division of labor [Smith, 1776] illustrates how breaking complex tasks into specialized functions dramatically increases productivity. Smith further observed that "the division of labor is limited by the extent of the market", meaning specialization increases as markets grow. This principle applies directly to models: as the demand for capabilities expands, we should expect greater specialization of models rather than continued focus on general-purpose systems.

David Ricardo's theory of comparative advantage [Ricardo, 1817] extends this insight, showing that even when one agent is superior at all tasks, the total output is maximized if agents specialize according to their relative strengths.

**Organizational Theory: Collective Decision-Making and Diversity.** Condorcet's Jury Theorem [Condorcet, 1785] provides mathematical proof that groups of independent decision-makers with better-than-random accuracy consistently outperform individuals, with reliability approaching certainty as group size grows. This applies directly to expert orchestration, where specialized judge models serve as an "expert jury" providing more reliable assessment than any single generalist model.

Lu Hong and Scott Page's diversity theorem [Hong and Page, 2004] extends this insight, proving that "groups of diverse problem solvers can outperform groups of high-ability problem solvers." Expert orchestration leverages this principle by maintaining diverse specialized models, each bringing distinct problem-solving approaches to user queries.

**Cognitive Science: Modularity and Distributed Intelligence.** Cognitive science provides compelling evidence that intelligence naturally emerges from specialized, interacting components rather than monolithic processors. Jerry Fodor's "Modularity of Mind" theory [Fodor, 1983] demonstrates that human cognition comprises domain-specific modules specialized for particular functions like language or vision, each operating with some independence from others. This modularity enables both efficiency and robustness—when one module fails, others continue functioning.

Building on this foundation, Marvin Minsky's "Society of Mind" theory [Minsky, 1986] offers a direct parallel to expert orchestration. Minsky proposed that intelligence emerges from "the interaction of many small, simple parts" without requiring a complex central controller: "a model of the human mind more like a democracy than a supercomputer." Recent AI research has validated this approach: Park et al. [2023] demonstrated that over a hundred specialized LLM agents working together can outperform any single model on complex tasks by collaborating and sharing information.

**Biological and Evolutionary Frameworks.** The evolution of multicellular life provides a compelling analogy for expert orchestration. Single-celled organisms function as generalists, handling all life processes internally. The transition to multicellularity involved cells specializing into different types (muscle, nerve, blood, etc.), dramatically increasing the organism's capabilities. As Rüffler et al. note, "division of labor among functionally specialized modules occurs at all levels of biological organization" and represents a major evolutionary trend because specialization enables higher performance [Rüffler et al., 2012].

**Routing as a Form of Democratic Algorithmic Institution.** Expert orchestration reflects key democratic values: participation, accountability, and distributed influence. Robert Dahl emphasizes that democracy depends on broad inclusion and equal ability to shape outcomes [Dahl, 2008], while Jürgen Habermas underscores the role of open, reasoned dialogue in legitimizing decisions [Habermas, 2015]. John Dewey sees democracy as collective problem-solving rooted in everyday association [Dewey and Rogers, 2012].

Expert orchestration echoes these ideals by lowering barriers for niche model creators, enabling a wider range of contributors to offer specialized capabilities. Through open evaluation and fair task routing, it promotes meaningful participation and healthy competition. Like the U.S. system of checks and balances, this distribution of influence helps prevent dominance by any single actor, fosters fairness, and supports systemic stability [Madison, 1788]. This pluralistic structure enables excellence across diverse domains and interests — an ideal at the heart of Walzer's argument for justice through distinct but coexisting spheres of merit [Walzer, 2008].

**Alignment and Safety Approaches.** The safety via debate framework [Irving et al., 2018] proposes training agents to engage in adversarial debates about questions, with a human or judge model determining which agent provides the most convincing answer. This approach uses multiple systems with potentially opposed viewpoints to surface flaws in each other's reasoning, improving the trustworthiness of answers. Expert orchestration naturally incorporates this debate-like structure through its judge models.

Christiano et al. [2018]'s Iterated Distillation and Amplification (IDA) alignment framework parallels expert orchestration principles. IDA starts with humans or simple models breaking complex tasks into smaller sub-questions, answering those questions, and then aggregating the answers. This decomposition approach is then distilled into a more efficient model, which is iteratively amplified through additional decomposition.

**Putting it all together.** Across economics, cognition, biology, and organizational theory, specialized coordinated systems consistently outperform monolithic designs for complex tasks. From Hayek's distributed knowledge to Minsky's society of mind to multicellular evolution, the pattern is clear: complex capabilities emerge through orchestrated interaction of specialized components, not through scaling generalist systems. Expert orchestration applies these proven principles to AI, creating systems that are more capable, transparent, and democratically governable than monolithic alternatives.

# 4 An Expert Orchestration Framework

The limitations of monolithic frontier LLMs call for alternative approaches. Here we outline the expert orchestration framework, a compelling vision designed to overcome several shortcomings.

**The Role of Judges.** At the core of expert orchestration are specialized models or systems that we term "judges". These judges are designed with a deep understanding of specific characteristics relevant to the evaluation of LLM outputs. Their primary role is to objectively assess these characteristics across a range of different LLMs. For example, there could be a judge specializing in evaluating the factual accuracy of an answer, another focused on determining its legality, a third assessing its adherence to ethical principles, and yet another dedicated to detecting the presence of hallucinations or biases. Currently, judges evaluate models based on their responses to user queries but other approaches are possible e.g. Kadavath et al. [2022] show that LLMs can often evaluate the validity of their own claims and predict which questions they'll be able to answer correctly.

The key attribute of these judges is their independence and objectivity, which are crucial for building trust and transparency in the evaluated characteristics of the various LLMs within the expert orchestration. By having different judges concentrate on distinct aspects, the framework enables a comprehensive evaluation of LLM outputs across multiple critical dimensions.

**The Role of Routers.** The second critical component of this expert orchestration framework is the "router." The router acts as an intelligent director, receiving user queries and making informed decisions about which LLM or combination of LLMs is best suited to address that specific query [Prem Blog, 2025b]. The routing decision is informed by the evaluations provided by the judges, as well as potentially by user-specified preferences regarding the desired characteristics of the response [Towards Data Science, 2025].

When a legal question arises, the system could transparently show that it is routing to a specialized legal model updated with recent case law rather than making users guess whether a generalist model's legal reasoning is reliable.

Routers can also take into account other factors such as the specialization of different models, their cost of operation, and their speed in generating responses [Prem Blog, 2025a]. A key advantage of this routing mechanism is its dynamic nature. The expert orchestration can readily adapt to the emergence of new and improved LLMs by incorporating them into the model set and utilizing the judges to assess their capabilities. This allows the system to continuously evolve and leverage the latest advancements. Routing techniques are being developed to optimize for cost, speed, and capability [Prem Blog, 2025b].

Recent advances in cost-aware model routing such as HybridLLM and CARROT [Ding et al., 2024, Somerstep et al., 2025] and adaptive MoE inference [Zhong et al., 2024], which dynamically select experts based on task relevance and efficiency tradeoffs, will support expert orchestration.

**Superior Performance.** There is much evidence in the machine learning literature [Hansen and Salamon, 1990, Dietterich, 2000, He et al., 2015, Devlin et al., 2018] that using model sets enables better performance than a single model. This is also true for expert orchestration, where judges are used to train router models for which LLM will perform well on each input query (pre-hoc methods), or to distinguish between the best results after calling multiple models (post-hoc methods).

Often, it is not obvious that pre-hoc methods may perform well for LLM applications; however there is mounting empirical evidence that pre-hoc routing is a cost-effective way to increase performance. The authors of the CARROT algorithm [Somerstep et al., 2025] show that you can train a model to predict the cost and quality of a model's generation from an input prompt, and then use this predictor at test time to select the best model for your cost or quality constraint. They show that CARROT is able to achieve higher scores

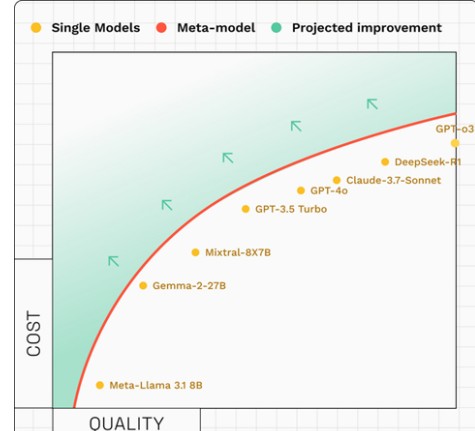

Figure 2: Judge and router "meta-models" out-perform any single model. Users select a point on the Quality / Cost pareto curve.

across the range of model costs than any single model on the RouterBENCH benchmark. Similarly, prompt-to-leaderboard [P2L; Frick et al. [2025]], which was trained on preference data gathered on the LMSys ChatBot Arena, was able to top the leaderboard by predicting which LLMs are preferred by users for different queries. Recent work on a universal model router [Shnitzer et al., 2024] was able to predict the performance of models unseen during training to achieve higher test-time scores on three separate benchmarks.

Post-hoc methods involve querying multiple LLMs and selecting (or creating) the best answer possible. For example, LLMBlender [Jiang et al., 2023] generates answers under many models, and then uses a fusion operation to generate the best answer. Simpler operations involving a judge might be to generate answers from many models and pick the best answer as scored by the judge. Though these methods are very powerful, they can also have high costs. However, other post-hoc ensembling methods can be considerably cheaper. FrugalGPT introduced a model cascade, where one model is run at a time, returning the first answer that exceeds a score threshold according to the judge. FrugalGPT was able to achieve significant performance improvements (up to 2.4x cost reduction while maintaining similar quality) at reduced costs [Chen et al., 2023]. Similarly, experiments from the RouterBench paper [Hu et al., 2024] showed that a similar cascade router outperformed single

models at multiple cost levels on 3 public benchmarks. It also shows that the quality of such a router is heavily dependent on the quality of the judge it utilizes.

Beyond cost and quality optimizations, expert orchestration can be used to trade off latency and memory constraints. Routing parallels the ideas behind Mixture-of-Experts (MoE) systems such as GLaM [Du et al., 2021] and Switch Transformers [Fedus et al., 2021], where routing to specialized components leads to improved performance-per-compute. These systems also outperform monoliths by activating the best model fragment per query. These claims are are also supported mathematically by Outrageously Large Neural Networks [Shazeer et al., 2017] that shows that sparse gating enables models to learn when and how to activate subcomponents for increased performance.

**Integration of Specialized Models.** Expert orchestration offers a significant advantage in its ability to seamlessly integrate specialized models into the broader ecosystem [Conclusion Intelligence, 2025]. Specialized models that demonstrates "best in class" performance in a certain domain, as evaluated the relevant judge, then the router can prioritize this model for such queries [Locaria, 2025]. This eliminates the current need for every innovator to develop a model that matches the broad capabilities of a frontier LLM to gain market share. Instead, innovators can focus their efforts on achieving excellence within a narrower scope [Dredze et al., 2024]. This fundamentally democratizes the process of model creation, fostering a vibrant community of specialist model innovators who can contribute valuable expertise to the expert orchestration.

We acknowledge that many specialized models will be derivatives of foundation models, at least in the current paradigm. However, this strengthens rather than undermines our argument. Expert orchestration enables the full value of foundation models to be realized through selective specialization, fine-tuning, and deployment. Rather than asking one model to perform optimally across all domains (an impossible task given parameter interference), orchestration leverages foundation capabilities while optimizing performance through specialized routing. This represents a more mature evolution of systems, similar to how early integrated computer systems eventually evolved into specialized components working in concert.

## 5   How Expert Orchestration Enhances Key Aspects of LLM Utility

Expert orchestration offers substantial enhancements across several key dimensions of LLM utility, leading to a more robust, user-centric, and responsible ecosystem.

**Increased Transparency and Trust.** A significant benefit of expert orchestration lies in its inherent ability to increase transparency and build trust in LLM outputs [IBM, 2025]. By employing dedicated, independent, and objective judges to evaluate specific characteristics of interest across a multitude of models, the framework provides users with a clearer understanding of the strengths and weaknesses of different LLMs in various domains [SmythOS, 2025]. This evaluation process makes the "thinking" more transparent compared to the often opaque decision-making of monolithic LLMs [PMC, 2025].

Expert orchestration parallels the rigorous, standardized evaluation practices found in safety-critical domains such as aviation, nuclear energy, and medical device manufacturing, where independent regulatory bodies and engineering frameworks are used to ensure system safety, reliability, and compliance [Leveson, 2016, Rushby, 1994, Storey, 1996]. Organizations like the Vector Institute and DNV already provide independent evaluations of models and vendors, highlighting the importance of this objective assessment [GlobeNewswire, 2025, DNV Group, 2025].

Recent work in Deep Interpretable Ensembles [Kook et al., 2022] and Judging the Judges: Evaluating Alignment and Vulnerabilities in LLMs-as-Judges [Thakur et al., 2024] shows the importance of exploring internal reasoning or decision factors to improve public trust.

**Selection of Judges Empowers Users.** The use of judges focusing on specific characteristics empowers users with greater control over responses [Phenx AI, 2025]. Consider a user who requires legally sound answers for critical actions. Expert orchestration allows this user to specify prioritized characteristics for individual requests, with the router directing queries to models best suited to provide aligned answers [Prem Blog, 2025a].

**Decomposing Requests Improves Alignment, Control, and Accuracy.** Expert Orchestration facilitates decomposing complex requests into manageable steps, such as planning followed by execution phases [Eyelevel.ai, 2025]. For planning, specialized project models can be utilized. The

resulting plan can be reviewed by "supervisor" models to enhance safety. Specialized costing models can estimate resources required, with execution steps delegated to diverse domain-specific models.

This decomposition provides natural monitoring points and reduces the "scope of control" of any single model, lessening reliance on potentially misaligned models and mitigating single points of failure. Untrustworthy models can be swapped out. Decomposition allows us to start developing robust control techniques now.

**Realigning Market Incentives Towards Specialization and Competition.** Expert orchestration fundamentally restructures market dynamics by eliminating both the transaction costs and competitive moats that make specialized models economically unviable [Varoquaux et al., 2025]. Currently, users face high switching costs when moving between different models for different tasks—learning new interfaces, managing multiple subscriptions, and remembering which model works best for what. These frictions make generalist models attractive despite inferior performance in specific domains. Simultaneously, incumbent companies build defensive moats by creating models that are "good enough" across many domains, making it hard for users to justify switching despite superior specialists existing. Expert orchestration destroys both barriers: it removes transaction costs through seamless automatic routing behind a unified interface, while eliminating defensive moats by automatically choosing the best model for each task. This makes it impossible to defend market position through convenience rather than capability—companies must continuously earn their position through specialized excellence.

Organizations implementing expert orchestration face structural incentives that naturally promote ecosystem health. To maximize routing accuracy and customer value, they must maintain comprehensive, objective evaluations of available models on a per domain basis, combating the proliferation of contaminated benchmarks in training datasets [Dodge et al., 2021, Deng et al., 2023] for general capabilities. In doing so, they are also economically motivated to continuously seek out and integrate the most effective specialized models. This creates sustainable market demand for niche innovators while incentivizing transparency: orchestration providers gain credibility through verifiable model evaluations rather than capability hoarding, creating competitive pressure toward better measurement and disclosure.

Furthermore, the organization is naturally driven to publish objective "leaderboards" that rank models based on their performance across various capability areas. This transparency provides a clear benchmark for innovators, who then only need to create a model that excels in a specific area to gain recognition and potential integration into an expert orchestration implementation. Expert orchestration stops the possibility of "best general model captures all value".

# 6 Research Directions and Open Challenges

Expert orchestration, while promising, presents several key research questions that warrant further investigation by the NeurIPS community. First, developing robust methodologies for evaluating models across diverse "thinking" characteristics beyond traditional metrics is essential, including bias [Team, 2025c], fairness [Team, 2025b], and hallucination detection [Team, 2025a].

Research is needed on utilizing multiple judges that reflect diverse user preferences to inform routing decisions, including aggregation methods and context-based weighting strategies. Deeply understanding model capabilities beyond simple benchmarking is necessary for optimal task matching.

Additional research directions include: (1) developing efficient and scalable routing algorithms that handle numerous models and complex preferences; (2) addressing the cold-start problem for new models with limited performance data; (3) exploring techniques for composing specialized models [Yang et al., 2024] to create more powerful capabilities; (4) studying broader ecosystem dynamics and impacts on competition and innovation; (5) applying dynamic model selection techniques [Brownlee, 2025] for adaptive routing; (6) developing theoretical models for when and how expert orchestration can improve performance such as with boosting; and (7) how to leverage ensemble methods [Chen et al., 2025] and cost-aware routing [Somerstep et al., 2025] to optimize performance and efficiency.

Research into different architectures for implementing judges – including fine-tuned specialized models, rule-based systems, and human evaluation integration – represents another critical area

for investigation. Together, these research directions will help realize the full potential of expert orchestration while addressing its current limitations.

## 7 Alternative Views and Considerations

As with any emerging framework, expert orchestration invites thoughtful critique and warrants a balanced evaluation. Several alternative perspectives surfaced during the development of this work:

**Market Consolidation Risks.** While expert orchestration aims to democratize participation in the LLM ecosystem, some argue it may ultimately reproduce existing inequalities. As specialist model development becomes lucrative, there is a risk that a few dominant actors could monopolize this space, particularly if orchestration itself becomes centralized or embedded within frontier AGI systems.

**Corporate Incentives and Public Benefit Structures.** This paper emphasizes misaligned incentives in frontier model development. However, critics note that some organizations, including OpenAI and Anthropic, operate as or are transitioning to public benefit corporations (PBCs). As such, they are legally permitted—and in some cases obligated—to prioritize societal welfare alongside shareholder value. This nuance complicates a purely profit-motivated critique.

**Latency and Cost Trade-offs.** The orchestration of multiple models introduces questions about computational efficiency. Decomposing queries and routing them through specialized evaluators and responders may increase latency or system overhead in certain cases. Nevertheless, as frontier models continue to grow in size and cost, the relative efficiency of using lightweight specialist models is likely to become increasingly favorable.

**Applicability Beyond Language.** Some readers may view expert orchestration as specific to LLMs. In practice, the framework generalizes to other modalities—including vision, speech, and multimodal systems—where specialized components can also enhance performance, transparency, and control.

**Sufficiency of Generalist Models.** A common objection is that current generalist models, particularly when augmented with tool use, are "good enough" for most applications. We contend that this view underestimates both the current limitations and long-term risks. Specialized systems consistently outperform generalists in high-stakes or knowledge-intensive domains, and running massive models for simple tasks remains inefficient. Crucially, expert orchestration offers structural benefits—transparent governance, distributed safety guarantees, and robust oversight—that generalist architectures and tool use alone cannot provide.

These perspectives highlight important avenues for ongoing reflection, implementation caution, and further research, which we believe strengthen rather than diminish the case for expert orchestration.

## 8 Conclusion: Towards a More Robust and Human-Aligned Future

The current dominance of monolithic frontier LLMs suffers from inherent limitations related to winner-take-all dynamics, misaligned safety incentives, barriers to entry for specialized models, limited user insight, and the inefficiencies of a one-size-fits-all approach.

Expert orchestration offers a compelling alternative that addresses these shortcomings by introducing an framework composed of specialized evaluation models ("judges") and intelligent routing systems ("routers"). This approach promises higher quality answers at a lower average cost by strategically leveraging the strengths of diverse models, including both frontier and specialized ones.

The framework enhances transparency and trust through independent evaluation, empowers users with granular control over desired characteristics, improves alignment and accuracy through request decomposition, and fosters a more democratic and open ecosystem. Moreover, an organization implementing expert orchestration has incentives naturally aligned with safety and transparency.

If AGI does not emerge suddenly from a single generalist system, an expert orchestration framework could achieve AGI earlier than general models. Our approach enables the development now of strong safeguards that decrease potential extinction-level threats.

By addressing many limitations of the current paradigm and offering a path towards a more user-centric and responsible future, expert orchestration holds significant promise for shaping the next generation of large language model applications.

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
