# OpenReview forum: "Beyond Monoliths: Expert Orchestration for More Capable, Democratic, and Safe Large Language Models"
_NeurIPS.cc/2025/Position_Paper_Track — Submitted to NeurIPS 2025 Position Paper Track_

### Official Review · Reviewer_muRF · 2025-08-07

**Significance:** 3
**Presentation:** 2
**Rating:** 3
**Confidence:** 4

**Summary:**

This paper argues for an "Expert Orchestration" framework as an alternative to the prevalent monolithic approach brought about by the development of LLMs by large corporations, which have the resources to train frontier LLMs that are general and good enough for multiple (all) tasks. The authors first provide critique about concentrating resources and models, focusing on the (i) monopolistic market dynamics brought about by the dominance of a few big companies (leading to more bias, speed to gain market share and less concerns about safety, fairness, societal equity) and the (ii) technical difficulties of building a know-it-all general monolithic model with add-ons tools such as RLHF, prompt engineering, with less understanding on how they work and "think" and control on their outputs. The authors then build an argument for expert orchestration from diverse fields (economics, organizational theory, cognitive science, biology) as well as democracy, alignment and safety aspects. The framework has two important roles: Judges (evaluator of experts wrt queries) and Routers (route queries to experts) and can lead to superior performance and provide benefits such as higher transparency, alignment, control, more competitive markets.

**Strengths:**

The topic that the authors have chosen to address is an important one and with the advent of LLMs that are owned and trained by large corporations and the community in general would prefer a more transparent and reproducible approach. The expert orchestration framework is presented as such and the authors use the pattern of specialization in other diverse fields such as market economics, knowledge distribution, organizational theory, cognitive science to support the need for an expert orchestration framework with diverse experts specializing in their fields over a single generalist model which can perform many tasks. The framework is straightforward with judges to provide evaluation scores across different metrics (which each judge is a specialist in) and routers to choose experts to address user queries.

**Weaknesses:**

1. The definition of the expert models is not clear in the paper. Are they just the judges and routers or task-specific? Expert or specialist models can be derived from LLMs using 'fixes' claimed by the authors, such as prompt engineering, RLHF. In this case, the development of LLMs and expert models are dependent and not mutually exclusive. The authors acknowledge this but claimed that "expert orchestration enables the full value of foundation models to be realized through selective specialization, fine-tuning, and deployment" and these are enabled by "fixes" that have been claimed to be detrimental.
2. The use of diverse fields to support the need for expert orchestration while interesting, may not fully reflect how foundational LLMs are trained and how human experts are developed. LLMs are trained with as much data (breadth and depth) as possible and experts LLMs (more depth with same breadth), compared to a human generalist (breadth) individuals and a human expert (depth with certain breadth). As such, what may work for other fields may not work for LLMs.
3. The idea of routing is not exclusive to expert orchestration and has already been used in training LLMs. It is not clear what is the value-add over current methods.

**Questions:**

1. Can you elaborate on how Figure 1 explains decomposition and routing? It is very abstract to me.
2. The figures are not referred in the text and seems to be arbitrarily placed.
3. How is the expert orchestration framework different from the agentic AI framework, which is also pushed by big corporations building LLMs? Is there any reason why it is not considered in the paper?
4. Are the benefits in Section 5 only achievable using the expert orchestration framework?
5. Will a hybrid approach work? Monolithic LLMs development with expert orchestration to improve them?

**Alternative Position:**

Yes, and alternative positions are trivial straw-man arguments

**Author Identification:**

No.

**Context:**

2

**Details Of Ethics Concerns:**

Minor concerns on the citations and references (some links are not updated and citations are wrong formats [line 387] and from blogs)

**Discussion:**

3

**Ethics:**

["NO or VERY MINOR ethics concerns only"]

**Position:**

Yes, the paper argues for or against a position related to machine learning.

**Support:**

2

**Thoroughness:**

4

---

### Official Review · Reviewer_4Va9 · 2025-08-10

**Significance:** 1
**Presentation:** 1
**Rating:** 3
**Confidence:** 4

**Summary:**

This paper considers that prevailing trajectory toward ever larger, more expensive generalist foundation models controlled by a handful of big companies limits innovation and constrains progress.

To argue this, they propose using an expert orchestration framework as an alternative approach to help research and development for LLMs.

**Strengths:**

The proposed framework is interesting.

The paper addresses important problems and highlights the benefits of using specialized models in integrated framework. It aggregates a wide spectrum of recent research (routing, MoE, continual learning, safety, bias) and even foundational economic/ethical texts.

Each identified challenge is paired with concrete, actionable strategies (e.g., cost‑aware routers for inference, balanced fine‑tuning to curb forgetting).

The paper also presents an interdisciplinary lens incorporating economic and ethical frameworks (e.g., Smith, Ricardo, Wu) frames technical choices within broader societal implications, encouraging responsible AI deployment.

**Weaknesses:**

The motivation of the paper is interesting. However, it should be considered the large scale foundation models and the proposed distributed orchestrated frameworks of smaller expert models have been both utilized by different companies, labs., researchers etc. for different tasks.

Therefore, it is not  clear how the proposed framework will lead a paradigm shift.

The manuscript focuses on benefits but does not discuss downsides (e.g., increased maintenance for routing, complexity of MoE gating). Practical concerns such as heterogeneous hardware, latency budgets, and regulatory compliance are omitted. That is, it is not clear how small research labs or individual researchers will be able to develop and employ this framework resolving the limitations imposed by the aforementioned big companies.

**Questions:**

What are the main downsides you anticipate for each recommendation (e.g., increased maintenance for routers, potential loss in robustness with MoE gating)?

Do you have any guidance on when a particular approach might be overkill for smaller deployments versus large‑scale production systems?

How do you foresee routing and MoE architectures performing on edge devices or heterogeneous hardware environments (e.g., GPU vs. TPU)?

Are there regulatory considerations (GDPR, HIPAA) that influence your choice of routing strategies or model partitioning?

Can you elaborate on the coordination logic between multiple specialized models in your proposed compound system?

**Alternative Position:**

Yes, and alternative positions are trivial straw-man arguments

**Author Identification:**

No.

**Context:**

2

**Discussion:**

2

**Ethics:**

["NO or VERY MINOR ethics concerns only"]

**Position:**

No, the paper argues that a specific technical approach is superior to other approaches.

**Support:**

2

**Thoroughness:**

4

---

### Official Review · Reviewer_m7e4 · 2025-08-17

**Significance:** 4
**Presentation:** 4
**Rating:** 6
**Confidence:** 3

**Summary:**

This paper analyzes the technical and structural problems of the mainstream monolithic LLM paradigm and proposes an alternative framework called Expert Orchestration. The authors argue that competition centered on large corporations not only limits innovation but also risks concentrating AI power. They further note the inherent weaknesses of single models. The proposed framework consists of two core components. First, independent Judge models evaluate LLM outputs using multi-dimensional criteria such as legality and ethics. Second, Router integrates these evaluations with user requirements to direct queries to the most suitable expert model. The authors claim this approach enhances performance, cost-efficiency, safety, and transparency.

**Strengths:**

- This is a very well-written paper and highly suitable for a position paper. It has a clear problem statement, justification, and future direction.

- Directly challenges the dominant monolithic LLM paradigm and proposes the concrete alternative of Expert Orchestration.

- It moves beyond simple critique by proposing specific components, such as 'Judge models' and 'Routers,' and provides a clear roadmap for the research community through its discussion of open research questions (Section 6).

- Grounded in reality by referencing existing technologies (e.g., Mixture-of-Experts, FrugalGPT), which supports feasibility.

**Weaknesses:**

- While the vision of a system where numerous models are fairly evaluated and seamlessly connected by objective 'Judge' models is compelling, it is also highly idealistic. The success of the entire framework could become critically dependent on the performance and objectivity of these Judge models.

- Implementation raises major engineering challenges, including API standardization, version control, and maintaining large-scale reliability.

**Questions:**

- How can the objectivity and reliability of Judge models be secured and validated?
- How can "expertise" be formally defined and measured to ensure that the Router makes optimal decisions when selecting among specialist models?

**Alternative Position:**

Yes, and alternative positions are well-considered and addressed by the argument

**Author Identification:**

No.

**Context:**

4

**Discussion:**

4

**Ethics:**

["NO or VERY MINOR ethics concerns only"]

**Position:**

Yes, the paper argues for or against a position related to machine learning.

**Support:**

3

**Thoroughness:**

4

---

### Note · Authors · 2025-08-28

**1-10 Additional Comments:**

The feedback was high quality and useful. The inability to discuss with reviewers their reviews was a bit painful. The lack of a revision process means any improvements to a position paper based on the feedback will only be visible to the next conference the paper is submitted to.

**1-11 Submit Again:**

Probably yes

**1-1 Submission Process:**

3

**1-2 Next Year:**

Probably.

**1-3 Future Development:**

No comment

**1-4 Interest:**

["Panel discussions with other position paper authors", "Mentorship programs for early-career researchers"]

**1-4 Other Interest:**

No comments

**1-5 Thoughtful:**

8

**1-6 Supportive:**

6

**1-7 Technical Aspects Versus Position:**

4

**1-8 Gate Keeping:**

6

**1-9 Camera Ready Changes:**

Absolutely. We will make several improvements based on the feedback.

**3-1 Review Response1:**

m7e4

**3-2 Reaction To Review1:**

Thank you for your feedback.

**Re: How can the objectivity and reliability of Judge models be secured and validated?**
Judge model reliability requires multi-layered validation: rigorous benchmarking against established datasets and expert assessments, competitive market dynamics that select for superior judging capabilities, structural independence where EO providers focus solely on orchestration rather than developing models (eliminating conflicts), and continuous A/B testing with outcome tracking. Poor performance serves as a clear indicator requiring immediate attention.

**Re: How can "expertise" be formally defined and measured for optimal Router decisions?**
Several approaches exist:

* Domain-specific benchmarks testing accuracy, reasoning, and task-appropriateness
* Model confidence scoring combined with historical performance data
* Market validation where poor expertise criteria get outcompeted by superior routing algorithms
* Mixture of experts literature inspiration inluding 1) "Mod-Squad" paper defines expertise via mutual information between task identity and expert choice - well-designed systems should have high mutual information, and 2) Pruning metrics where removing experts causes benchmark performance drops in optimal systems

**3-3 Review Response2:**

4Va9

**3-4 Reaction To Review2:**

Thank you for your feedback. Most points address implementation details beyond this position paper's scope. While some specifics remain unimplemented, we believe they're all technically solvable and don't fundamentally undermine our core thesis.

**Re: What are the main downsides for each recommendation?**
EO frameworks introduce operational overhead alongside their benefits. Maintenance requirements include regular router retraining as model landscapes evolve, developing partially private evaluation datasets to prevent gaming by model creators, and managing increasingly granular capability classifications. The trade-off between added complexity and improved performance/cost efficiency will determine adoption rates.

**Re: Guidance on when approaches might be overkill vs. appropriate?**
Simple heuristics apply here: EO is likely overkill when cost isn't a primary concern, queries are generic and broad, volumes are relatively low, and frontier model performance already suffices. EO becomes valuable when cost optimization matters, query volumes are high, and optimal results are critical—particularly in high-risk domains requiring specialized expertise like law or medicine. The precise threshold will clarify as EO implementations mature.

**Re: Routing/MoE performance on edge devices or heterogeneous hardware?**
The framework supports specialist models across diverse hardware configurations by matching model requirements to available compute resources. Models will document their capabilities and hardware needs, enabling informed deployment decisions and appropriate query escalation when edge resources prove insufficient.

**Re: Regulatory considerations influencing routing strategies?**
This is an active research area requiring further investigation.

**Re: Coordination logic between specialized models?**
This complex topic merits detailed discussion beyond this paper's scope. We're currently testing multiple approaches and welcome post-publication dialogue.

**3-5 Review Response3:**

muRF

**3-6 Reaction To Review3:**

Thank you for this insightful meta-question. After extensive consideration, we believe EO represents the most viable path to achieving these benefits, though we welcome other proposals that could deliver similar outcomes.

**Re: How is expert orchestration different from agentic AI?**
Agentic AI uses LLMs as core reasoning engines with tool-calling and multi-turn conversations. EO complements this by intelligently routing to the most appropriate model for each task, enhancing rather than competing with agentic frameworks.

On an abstract level, if agentic systems include routers and orchestrators as tools, then EO falls under this broad definition—like how biology includes enzymatic processes. However, we see value in EO's sharper definition, enabling optimized algorithms and metrics.

**Re: Are Section 5's benefits only achievable through expert orchestration?**
We don't see alternative pathways. Current monolithic development won't deliver them, while EO creates a sustainable ecosystem with aligned incentives, fostering broader innovation. The framework's self-reinforcing nature—where financial and safety motivations align—uniquely positions it to achieve these outcomes.

**Re: Will a hybrid approach work? Monolithic LLM development with EO?**
Yes. Our EO framework includes frontier monolithic LLMs, routing to them when optimal. Some providers already route internally to smaller models for efficiency—a subset of our vision. However, they're unlikely to route to external specialists as this undermines their competitive positioning. Their business models inherently oppose true EO implementation.

**Re: Can you elaborate on Figure 1's decomposition and routing explanation?**
It is abstract. We'll replace it with a more informative version!

**Re: The figures aren't referenced in text and seem arbitrarily placed.**
Thank you for the feedback. We'll resolve this in the next version.

---

### Meta-Review · Area_Chair_M398 · 2025-09-07

**Rating:** 4
**Confidence:** 2

**Strengths:**

- **Clear Positioning**: The paper defines a specific problem (monolithic LLM dominance) and offers a concrete alternative: Expert‑Orchestration with specialized models.

- **Concrete Architecture**: The paper introduces tangible components—Judge Models for evaluation and Routers for task assignment—moving beyond high‑level critique to actionable design.

- **Roadmap & Open Questions**: Section 6 outlines explicit research directions and challenges, guiding the community toward future work.

- **Grounded Feasibility**: References existing systems (Mixture‑of‑Experts, FrugalGPT) demonstrate that the proposed framework can be built with current technologies.

- **Interdisciplinary Foundation**: The argument draws on economic theory, organizational science, and cognitive models to justify specialization over generalization.

- **Ethical & Economic Lens**: By incorporating discussions of bias, safety, cost‑efficiency, and societal impact, the paper frames technical choices within broader responsible‑AI considerations.

- **Comprehensive Literature Coverage**: It synthesizes recent research on routing, MoE, continual learning, and foundational texts, establishing a solid knowledge base for the proposed framework.

**Weaknesses:**

- **Idealistic Core Dependence**: The framework hinges on Judge models whose performance/objectivity directly determines overall success; no robustness analysis is provided.

- **Engineering Burden**: Practical challenges (API standardisation, versioning, large‑scale reliability) are acknowledged but not addressed or scoped.

- **Uncertain Paradigm Shift**: Similar distributed expert systems already exist; the manuscript fails to demonstrate how its approach would materially shift practice.

- **Missing Trade‑offs**: No discussion of downsides such as increased routing maintenance, MoE gating complexity, heterogeneous hardware, latency constraints, or regulatory hurdles.

- **Ambiguous Expert Definition**: The paper does not clearly distinguish expert models from judges/routers; it conflates fine‑tuned LLMs with truly specialized models, undermining the claimed benefits.

- **Misapplied Interdisciplinary Analogy**: Arguments drawn from economics or cognitive science may not translate to LLM training dynamics (breadth vs. depth), limiting relevance.

- **Limited Novelty of Routing**: Routing has precedent in MoE and LLM training; the paper does not articulate a clear added value over existing routing mechanisms.

**Questions:**

The Author Survey has answered the questions raised by the reviewers. No further questions need to be articulated.

**Ethics:**

No.

**Thoroughness:**

2

---

### Decision · Program_Chairs · 2025-09-26

Reject